
# A 3-D evaluation of the MACC reanalysis dust product over Europe, Northern Africa and Middle East using CALIOP/CALIPSO dust satellite observations

Aristeidis K. Georgoulias[1,2,3*], Athanasios Tsikerdekis[4], Vassilis Amiridis[5], Eleni Marinou[5,6], Angela Benedetti[7], Prodromos Zanis[4], Georgia Alexandri[1], Konstantinos A. Kourtidis[1], Jos Lelieveld[2,3]

[1]Department of Environmental Engineering, School of Engineering, Democritus University of Thrace, Xanthi, Greece

[2]Max Planck Institute for Chemistry, Mainz, Germany

[3]Energy, Environment and Water Research Center, The Cyprus Institute, Nicosia, Cyprus

[4]Department of Meteorology and Climatology, School of Geology, Aristotle University of Thessaloniki, Thessaloniki, Greece

[5]Institute for Astronomy, Astrophysics, Space Application and Remote Sensing, National Observatory of Athens, Athens, Greece

[6]Laboratory of Atmospheric Physics, Department of Physics, Aristotle University of Thessaloniki, Thessaloniki, Greece

[7]European Centre for Medium-Range Weather Forecasts, Reading, UK

*Correspondence to*: Aristeidis K. Georgoulias (argeor@env.duth.gr)

**Abstract.** The MACC reanalysis dust product is evaluated over Europe, Northern Africa and Middle East using the EARLINET-optimized CALIOP/CALIPSO pure dust satellite-based product LIVAS (2007-2012). MACC dust optical depth at 550 nm ($DOD_{550}$) data are compared against LIVAS $DOD_{532}$ observations. As only natural aerosol (dust and sea salt) profiles are available in MACC, here we focus on layers above 1 km a.s.l. to diminish the influence of sea salt particles that typically reside at low heights. So, MACC natural aerosol extinction coefficient profiles at 550 nm are compared against dust extinction coefficient profiles at 532 nm from LIVAS assuming that the MACC natural aerosol profile data can be similar to the dust profile data, especially over pure continental regions. It is shown that the reanalysis data are capable of capturing the major dust hot spots in the area as the MACC $DOD_{550}$ patterns are close to the LIVAS $DOD_{532}$ patterns throughout the year. MACC overestimates DOD for regions with low dust loadings and underestimates DOD for regions with high dust loadings where DOD exceeds ~0.3. The mean bias between the MACC and LIVAS DOD is 0.025 (~25%) over the whole domain. Both MACC and LIVAS capture the summer and spring high dust loadings, especially over Northern Africa and Middle East, and exhibit similar monthly structures despite the biases. In this study, dust extinction coefficient patterns are reported at four layers (layer 1: 1200-3000 m a.s.l., layer 2: 3000-4800 m a.s.l., layer 3: 4800-6600 m a.s.l. and layer 4: 6600-8400 m a.s.l.). The MACC and LIVAS extinction coefficient patterns are similar over areas characterized by high dust loadings for the first 3 layers. Within layer 4, MACC overestimates extinction coefficients consistently throughout the year over the whole domain. MACC overestimates extinction coefficients compared to LIVAS over regions away from the major dust



sources while over regions close to the dust sources (Sahara and Middle East) underestimates strongly only for heights below ~3-5 km a.s.l. depending on the period of the year. In general, it is shown that dust loadings appear over remote regions and at heights up to 9 km a.s.l. in MACC contrary to LIVAS. This could be due to the model performance and parameterizations of emissions and other processes, due to the assimilation of satellite aerosol measurements over dark surfaces only or due to
a possible enhancement of aerosols by the MACC assimilation system.

# 1 Introduction

Aeolian dust is mainly produced naturally by disintegration of soil aggregates over deserted, arid and semi-arid areas. The amount of dust emitted into the atmosphere depends on surface wind speed and on factors such as soil texture, soil moisture and vegetation cover (IPCC, 2013). Dust is also produced locally from anthropogenic activities (e.g. manufacturing,
construction, mining, agricultural activities, herding livestock, off-road vehicles and warfare) (Zender et al., 2004). On a global scale, it has been estimated that natural sources account for ~75% and anthropogenic sources for ~25% of the dust emissions (Ginux et al., 2012).

Climate and the biogeochemical cycles of various ecosystems (terrestrial and oceanic) are affected significantly by dust (Cuevas et al., 2015 and references therein). Dust modulates the radiative budget in the Earth-Atmosphere system directly
and indirectly by changing the microphysical and macrophysical properties of clouds acting as cloud condensation nuclei (CCN) and ice nuclei (IN) (IPCC, 2013 and references therein). Also, mineral dust affects human health in various ways (e.g. causing allergies, respiratory problems, eye infections, cardiopulmonary diseases, lung cancer) being related to more than 400000 premature deaths per year on a global basis (Giannadaki et al. 2014; Lelieveld et al., 2015). The deposition of dust can also reduce crop yields and lead to livestock losses (Sivakumar, 2005). Strong episodic dust events hamper visibility
affecting this way air and road transportation (De Villiers and Van Heerden, 2007) while the deposition of dust on solar panels affects their energy production efficiency (Beattie et al., 2012).

According to Ginux et al. (2012) 55% of the global dust emissions originate in North Africa, with only 8% being anthropogenic (mostly from the Sahel region). Significant amounts of dust are transported over Europe from the Sahara Desert and the Arabian Peninsula after crossing the Mediterranean Sea (Georgoulias et al., 2016a and references therein) and
also from smaller local sources (see Alastuey et al., 2016). Taking into account the determinant role of dust in processes related to weather and climate, human health and the economy, it is obvious that simulating adequately the amount of dust and its optical properties over the region is essential.

The World Meteorological Organization (WMO) acknowledged this by establishing the Sand and Dust Storm Warning Advisory and Assessment System (SDS-WAS) in 2007 that provides daily dust forecasts from more than 15 organizations
for different geographic regions (currently for the Northern Africa-Middle East-European region and for Asia). Among other global and regional models the MACC (Monitoring Atmospheric Composition and Climate) aerosol system of the European Centre for Medium-range Weather Forecasts (ECMWF) (Morcrette et al., 2009; Benedetti et al., 2009) provides 3-day dust



forecasts (optical depth and surface concentration) on a daily basis. The aerosol forecasts are produced using the same system that was operated for the production of the multiyear (2003-2012) MACC reanalysis. The MACC reanalysis was developed within the framework of GMES (Global Monitoring for Environment and Security) and a series of MACC projects funded by the European Union and coordinated by ECMWF (http://www.gmes-atmosphere.eu/about/project/). The

MACC activities are now carried on under CAMS (Copernicus Atmosphere Monitoring Service) (Eskes et al., 2015).

Upon its release the MACC reanalysis aerosol product has been used in a many studies at a global and regional level. For example, it has been used in global estimates of the direct and indirect aerosol radiative effect (Bellouin et al., 2013), to study the sensitivity of clouds to aerosol loads and types over the oceans (Andersen et al., 2016), to constrain the influence of aerosols on cloud coverage (Gryspeerdt et al., 2016), to build regional climatologies in conjunction with satellite data (e.g.

Nabat et al., 2013, Georgoulias et al., 2016a), as input for the production and evaluation of satellite-based surface solar radiation products (Mueller et al., 2015; Alexandri et al., 2017), to support reports on the current state of the climate (Benedetti et al., 2014), etc.

Specifically in dust oriented studies, MACC forecasts have been used in conjunction with measurements from dropsondes and lidars onboard aircrafts, ships and satellites (Cloud-Aerosol Lidar with Orthogonal Polarization onboard Cloud-Aerosol

Lidar and Infrared Pathfinder Satellite Observations - CALIOP/CALIPSO) to study the long-range transport of Saharan dust across the Atlantic within the framework of the Saharan Aerosol Long-range Transport and Aerosol-Cloud-interaction Experiment (SALTRACE) campaign in spring and summer 2013 (Chouza et al., 2016; Ansmann et al., 2017). In these studies forecast fields were used instead of analyses (MACC reanalysis data stop in 2012) focusing on the total aerosol optical depth (AOD) and extinction coefficients rather than on dust. Cuevas et al. (2015) evaluated the MACC reanalysis

dust product over Northern Africa and Middle East for two years (2007-2008) using ground and satellite-based measurements. Their comparisons focused on specific sites (AERONET sunphotometers, lidars and CALIOP/CALIPSO observations) while for spatial evaluations they utilized total AOD satellite data from passive sensors such as MODIS, MISR and OMI. Marinou et al. (2017) furthermore performed a first comparison of an optimized CALIOP/CALIPSO pure dust optical depth (DOD) patterns with MACC reanalysis DOD patterns; however, a detailed 3-dimensional spatiotemporal

evaluation of the MACC dust product is still missing.

In this study we advance for the first time to a 3-D (optical depths and profiles) evaluation of the MACC reanalysis dust product over the Europe - Northern Africa - Middle East domain [13$^{\circ}$ N-60$^{\circ}$ N, 40$^{\circ}$ W-70$^{\circ}$ E] using the EARLINET-optimized CALIOP/CALIPSO pure dust satellite-based product for the time period 2007-2012. It has to be highlighted that this is an independent observational product as it is not included in the MACC assimilation procedure. Details about the

datasets used for the evaluation along with a description of the methodology followed are given in Sect. 2. The results from the evaluation procedure are presented in Sect. 3, while in the end of the paper the main findings and conclusions of this research are summarized.



# 2 Data and methods

## 2.1 MACC reanalysis data

In this work, two MACC reanalysis datasets, one characterizing the columnar dust load and one indicative of the dust profile in the atmosphere, are evaluated for a six-years period spanning the period from 2007 to 2012. 3-hourly dust optical depth data at 550 nm ($DOD_{550}$) and natural aerosol (dust and sea salt) optical depth at 550 nm ($AOD_{550}$) profiles are available from MACC. The spatial resolution of the data is ~ 78 km x 78 km with 60 vertical levels from the surface up to 0.1 hPa. It is obvious that the profile data as given by MACC are affected by a sea salt component; however, if used properly one can get an insight into the ability of MACC to simulate the dust profiles. In addition to the optical depth data, geopotential data (in $m^2/s^2$) from MACC are also used in order to calculate the physical height of each model layer which is necessary for the comparison of the profile data with the satellite data as it is shown below.

The MACC reanalysis data used here are produced using the aerosol analysis and forecast system of ECMWF. This consists of a forward model (Morcrette et al., 2009) and a data-assimilation module (Benedetti et al., 2009). The MACC forecasting system assimilates, among other observational data (Eskes et al., 2014), $AOD_{550}$ measurements from the two MODIS sensors aboard Terra and Aqua through a 4D-Var assimilation algorithm to produce the aerosol analysis. The assimilation improves the representation of aerosols as shown in previous studies (see Benedetti et al., 2009; Mangold et al., 2011). The MACC aerosol system accounts for a total of five aerosol species, mineral dust, sea salt, sulfates, black carbon and organic matter. Three different size bins are used for mineral dust (0.03-0.55, 0.55-0.9 and 0.9-20 microns) and sea salt particles (0.03-0.5, 0.5-5 and 5-20 microns). Black carbon and organic material are distributed to a hydrophilic and a hydrophobic mode. Sea salt emissions are given as a function of surface wind speed (Guelle et al., 2001; Schulz et al., 2004). Dust emissions are given as a function of surface wind speed, soil moisture, surface albedo and land cover following Ginoux et al. (2001). The emissions of the other species are taken from inventories (e.g. SPEW, EDGAR) while a climatology is used for stratospheric aerosols.

## 2.2 LIVAS CALIOP/CALIPSO data

For the evaluation of the MACC reanalysis data, dust optical depth at 532 nm ($DOD_{532}$) and dust extinction coefficients at 532 nm (in $km^{-1}$) from CALIOP/CALIPSO are used. The horizontal resolution of the data is $1^o$ x $1^o$ and the extinction coefficient retrievals are available at 399 predefined heights which characterize a layer of ~60 m for altitudes below ~20 km and ~180 m for higher altitudes. As CALIPSO flies at a 705 km altitude sun synchronous polar orbit with a 16 day repeat cycle there are 1-3 measurements available per grid cell on a monthly basis. The satellite data utilized in this work have been produced using an EARLINET-optimized retrieval scheme that was developed within the framework of the LIVAS (LIdar climatology of Vertical Aerosol Structure for space-based lidar simulation studies) project (Amiridis et al., 2015). More specifically the pure dust LIVAS product is used (see Amiridis et al., 2013; Marinou et al., 2017). This product is corrected





for the dust LR value which is specific for the region we focus in this study, based on multi-year measurements performed by the ground-based lidar stations of the EARLINET (European Aerosol Research Lidar Network). These region-specific LRs are equal to 55±5 sr for the North Africa - Europe domain (Tesche et al., 2009, 2011; Groß et al., 2011, 2015) and equal to 40±5 sr for Middle East and central Asia at longitudes further east than 30°E (Mamouri et al., 2013; Nisantzi et al., 2015;

Hofer et al., 2017). The correction leads to an $AOD_{532}$ absolute bias of ~-0.03 compared to spatially and temporally collocated AERONET observations above Europe and North Africa while the corresponding biases for the standard CALIPSO product are much higher (~-0.10) (Amiridis et al., 2013). The bias is lower (~-0.02) when compared against spatially and temporally collocated MODIS satellite data. In addition, the use of a new methodology for the calculation of the pure dust extinction from dust mixtures and an averaging scheme that includes zero extinction values for the non-dust

aerosol types allow for further improvement of the LIVAS pure dust product (Amiridis et al., 2013).

## 2.3 Spatial and temporal collocation of the datasets

The DOD and profile datasets from MACC reanalysis have to be processed properly prior to the comparison with the LIVAS data. Generally, it is much more straightforward to evaluate the MACC columnar dataset. It has to be mentioned that while the MACC reanalysis data are available on a 3-hourly basis, the LIVAS data used here are available as monthly means.

However, the exact overpass date and time of the retrievals used for the calculation of the monthly data is given which allows for the temporal collocation of the two datasets. The MACC $DOD_{550}$ data are first brought to the LIVAS 1° x 1° grid using bilinear interpolation and then only the MACC values closer to the to the LIVAS $DOD_{532}$ retrieval time are chosen. Finally, the MACC $DOD_{550}$ data are averaged on a monthly basis and can be evaluated against the LIVAS data (see Fig. 1a for the whole procedure).

Much more effort is needed to bring the MACC reanalysis and the satellite-based profile data in a format suitable for comparison (see Cuevas et al., 2015 and Chouza et al., 2016 for previous efforts) prior to the horizontal and temporal collocation of the datasets. As the MACC reanalysis offers only natural (dust and sea salt) $AOD_{550}$ fields (unitless) for each one of the 60 MACC layers and the LIVAS data include extinction coefficients (in km$^{-1}$) at 399 heights it is obvious that the two datasets are not directly comparable. Similar problems may emerge when evaluating simulations from other global or

regional climate models. The method proposed here (see Fig. 1b for details) is a generic one and could be applied in future model evaluation studies. First, the MACC reanalysis $AOD_{550}$ profiles are converted to extinction coefficients at 550 nm by dividing the given MACC geopotential fields with the gravity acceleration to obtain the physical layer heights. From the physical layer heights (upper layer minus lower layer physical height) the physical depth of each MACC layer is calculated. Finally, the $AOD_{550}$ of each layer is divided by the layer's physical depth in order to calculate the natural aerosol extinction

coefficient (in km$^{-1}$) that characterizes the whole layer. Then the MACC profiles are linearly interpolated to the 399 LIVAS levels, and similar to the case of DOD, the MACC profile data are brought to the LIVAS 1° x 1° grid using bilinear interpolation. Only the MACC values closer to the to the LIVAS retrieval time are chosen and the MACC reanalysis natural





aerosol extinction coefficients for the 399 LIVAS levels are averaged on a monthly basis. To obtain more robust statistics, both the MACC and the LIVAS 399-level data are finally averaged vertically within a set of selected layers each one having a depth of 300 m (see also Cuevas et al. 2015). The evaluation procedure is implemented for 29 (300-meter) layers covering the troposphere from 300 m (first layer centered at 450 m) up to 9 km (last layer centered at 8.85 km). It needs to be

reiterated that the profile data as given by MACC are contaminated with a sea salt component; however, if used properly one can get an insight into the ability of MACC to simulate the dust profiles. Sea salt particles in the area are mostly accumulated within the marine boundary layer, generally at heights below 1 km (see Nabat et al., 2013); hence, sea salt is expected to have an impact only at the lower levels of the natural aerosol profiles. Therefore, in this work we focus on layers higher than 1 km. The extinction coefficient patterns (MACC, LIVAS and their difference) presented in this work are calculated by

averaging vertically over four 1800-meter layers (layer 1: 1200-3000 m, layer 2: 3000-4800 m, layer 3: 4800-6600 m and layer 4: 6600-8400 m). The first layer starts from 1200 m (> 1 km) in order to diminish the contamination of the extinction coefficients from sea salt particles as discussed above. Hence, we refer to MACC dust extinction coefficients hereafter and not natural aerosol extinction coefficients. One should keep in mind however that the dust extinction coefficients used here are still contaminated with a sea salt component at some degree especially over the sea and regions close to the coasts while

our results should be considered more robust over pure continental regions.

## 2.4 Evaluation procedure

The MACC reanalysis dust product evaluation procedure comprises different steps. First the annual MACC $DOD_{550}$ patterns are compared against observations from the LIVAS $DOD_{532}$. The Europe - Northern Africa - Middle East (EU) domain and nine sub-regions are used for the generalization of the results. The sub-regions of interest are: Central Europe (CE), Eastern

Europe (EE), South-western Europe (SWE), Central Mediterranean (CM), Eastern Mediterranean (EM), Atlantic Ocean (ATL), Central-western Sahara (CWSah), Eastern Sahara (ESah) and Middle East (ME). The average MACC and LIVAS DODs for the period 2007-2012 along with the mean bias (MB), the normalized mean bias (NMB), the root mean squared error (RMS error), the correlation coefficient (R), the slope (a) and the intercept (b) of the MACC-LIVAS linear regression line are reported for EU and for each sub-region of interest. Then the seasonal MACC $DOD_{550}$ patterns are compared against

$DOD_{532}$ observations from LIVAS for winter (December-January-February: DJF), spring (March-April-May: MAM), summer (June-July-August: JJA) and autumn (September-October-November: SON). In addition the monthly variability of the MACC and LIVAS DOD and their difference is given. As a next step, the annual patterns of the MACC dust extinction coefficient at 550 nm patterns are compared against the LIVAS dust extinction coefficient at 532 nm patterns for the four layers defined in the previous paragraph. The average MACC and LIVAS extinction coefficients for the period 2007-2012

are reported for each layer for EU and for each sub-region of interest along with MB, NMB, RMS error, R, a and b. 300-m resolution profiles of the MACC and LIVAS dust extinction coefficients are also compared. Finally, the seasonal patterns of




the difference between the MACC and LIVAS dust extinction coefficients are presented for the four layers along with the monthly variability of the difference between the MACC and LIVAS 300-m resolution profiles for all the regions of interest.

# 3 Results and Discussion

## 3.1 Evaluation of the columnar MACC reanalysis dust dataset

### 3.1.1 Annual dust optical depth patterns

In this section, the evaluation of the MACC columnar dust load is presented. As shown in Figs. 2a, b the annual MACC $DOD_{550}$ patterns are close to the LIVAS $DOD_{532}$ ones showing that the reanalysis can capture the observed spatial distribution and the major dust hot spots in the area. However there are some discrepancies in the magnitude as indicated in Fig. 2c. The MACC-LIVAS DOD MB patterns shown in Fig. 2c are characterized by a general overestimation of DOD by MACC over continental Europe, over parts of Turkey and Iran, and over the sea (Atlantic Ocean, Mediterranean and Arabian Sea). The overestimation is larger over the region situated on the west of the Caspian Sea and the region of the eastern Sahara. On the other hand, MACC underestimates DOD significantly over the region of central and western Sahara and over part of the Middle East. From the comparison of Figs. 2a, b and c it can be seen that DOD is underestimated by MACC for areas where DOD exceeds ~0.3. Indeed, by applying a frequency distribution analysis on the whole MACC and LIVAS data record (see Fig. 2d) it is shown that on average the bias becomes negative when DOD becomes higher than 0.28. It has to be highlighted here that ~90% of the LIVAS $DOD_{532}$ values are below this critical value (see Fig. 2d), hence, the underestimations are connected to a large degree with source areas and episodic dust events.

Over the whole domain (EU), the MB between the MACC and LIVAS DOD is 0.025 and the NMB is ~25% with an RMS error of 0.115. The correlation coefficient (R) of the linear regression (y=0.562x+0.068) between MACC and LIVAS DOD is 0.76 (for details see Table 1). Up to a DOD value of ~0.3 the MACC and LIVAS products are characterized by a strong linear correlation with a slope close to 1 (y=0.986x+0.046 with R=0.76); however, there is no significant correlation for values higher than ~0.3, LIVAS exhibiting much higher DOD values than MACC (y=0.176x+0.269 with R=0.33). A similar situation is observed over sub-regions high DODs near the major dust sources such as CWSah, ESah and ME and less over the transitional (from dusty to clean conditions) sub-regions such as ATL, SWE, CM and EM. The sub-region with the highest MACC-LIVAS correlation (higher R value and slope closer to one) is ATL (y=0.713x+0.037 with R=0.87) and the sub-region with the lowest MACC-LIVAS correlation is CE (y=0.300x+0.029 with R=0.40). The slope and the intercept of the linear regression line, the correlation coefficient R, the MACC-LIVAS MB, NMB and the RMS error, the MACC and LIVAS mean DOD levels for all the sub-regions of interest are given in Table 1.

It is concluded from the two paragraphs above that in general MACC overestimates DOD for regions with low dust loadings and underestimates DOD for regions with high DOD loadings. Similar results were shown in Amiridis et al. (2013) and





Tsikerdekis et al. (2017) where BSCDREAM8b and RegCM4 dust simulations were compared against CALIOP/CALIPSO satellite observations. Many reasons could be responsible for these overestimations/underestimations. First of all it might be related to the model itself (e.g. parameterization of dust emissions, the wind velocity, the distribution of dust particles in different bins, the dry and wet deposition, the convection scheme which is used, etc.). For example if the model

overestimates the fine mode dust particles, the lifetime of dust in the air would increase leading to the transport of particles away from the sources and at greater height levels. However, as discussed in Ansmann et al. (2017), the uncertainties stemming from the complex parameterizations used by the model make it difficult to reach a solid conclusion about the observed overestimations and underestimations.

Another reason for the underestimation of DOD close to the major dust sources in the area could be the assimilation of

$AOD_{550}$ measurements only from the MODIS/Terra and MODIS/Aqua Dark Target (DT) product which does not include observations over bright surfaces such as deserts, arid and semi-arid regions. The overestimation of dust away from the sources (MACC DOD never gets a zero value even over remote oceanic regions) might also be related to the assimilation procedure. The control variable of the assimilation is the total aerosol mixing ratio calculated by adding the mixing ratios of all species. The $AOD_{550}$ is calculated from the single species, summed, integrated and then compared to the observations.

Through the 4D-Var assimilation algorithm increments in total aerosol mixing ratio are obtained. Those increments are redistributed to all species proportionally to their fractional contribution to the total mass. It has to be noted that the model does not take into account ammonium nitrate aerosols which represents a large component over the greater European area (Giordano et al., 2015). As a result the model will most of the time underestimate AOD relative to the observations and hence the assimilation system will tend to increase the other aerosol components to give the correct AOD overall. Probably,

the system allows the presence of dust even at tiny concentrations and so dust always receives a small contribution during the assimilation even when there should be no dust in the atmosphere.

One more parameter contributing to the differences observed between the model and the observations is probably the limitation of CALIPSO to detect aerosol layers with signals lower that the satellite's signal-to-noise ratio (SNR) (Winker et al., 2013). In particular, in heights where the CALIPSO SNR is higher that the signal of the layer, the area is characterized as

clear air, and a value of 0 km$^{-1}$ is set. The detection thresholds are defined in terms of 532 nm scattering ratio and are adjusted according to altitude, solar background illumination and averaging resolution (Vaughan et al., 2009). As higher thresholds are used during daytime than at night (because SNR is reduced by solar background illumination), weakly scattering layers which are detected at night may be missed during daytime. Typical values of CALIPSO layer thresholds in dust observations are 0.04±0.02 km$^{-1}$ during daytime and 0.008±0.003 km$^{-1}$ during nighttime. Indicatively, Kim et al. (2017)

found a global mean undetected aerosol layer with an AOD of 0.031±0.052 after comparing 2 years of CALIPSO and MODIS AODs. These undetected layers are expected to affect more the higher altitudes in the CALIPSO product.



### 3.1.2 Dust optical depth seasonal variability

In this section, the seasonal variability of the MACC, the LIVAS DODs and their differences are discussed. The seasonal patterns of MACC $DOD_{550}$ and LIVAS $DOD_{532}$ along with the MACC-LIVAS MB patterns are presented in Fig. 3 while the corresponding monthly variabilities per region of interest are shown in Fig. 4. A number of studies using passive and active

satellite-based observations have revealed the spatiotemporal variability of dust and its pathways over the greater Mediterranean area during the last two decades (e.g. Moulin et al., 1998; Prospero et al., 2002; Barnaba and Gobbi, 2004; Antoine and Nobileau, 2006; Gkikas et al., 2009, 2013, 2016; Israelevich et al., 2012; Ginoux et al., 2012; Pey et al., 2013; Varga et al., 2014; Georgoulias et al., 2016a,b; Tsikerdekis et al., 2017; Marinou et al., 2017). It is well known today that over western Mediterranean dust peaks in summer, over eastern Mediterranean in spring while central Mediterranean is a

transitional region with high dust loadings throughout summer and spring. Dust concentrations over western Europe peak in summer while over central and eastern Europe are higher during spring and summer than during the rest of the year. The seasonal variability of dust depends mostly on the seasonality of the emissions over the source areas and the dominating wind patterns. Over the western Sahara the dust emissions peak in summer, over the eastern part of the desert in spring and over the Middle East dust activity peaks in late spring and summer (for details see in the studies given above and the

references therein). As shown in Fig. 3, the MACC reanalysis DODs (Figs. 3a, d, g and j) exhibit a similar spatial variability with the LIVAS DODs (Figs. 3b, e, h and k) for all the seasons. The well documented high dust loadings during summer and spring over the whole domain, especially over Northern Africa and Middle East, are depicted by both the MACC and the LIVAS DODs. However, in line with Fig. 2 (annual patterns), for areas away from the sources, MACC overestimates DOD during spring, summer and autumn and to a lesser extent in winter. This is more profound in the MACC-LIVAS MB patterns

shown in Figs. 3c, f, i and l. The MB values over these areas are up to 0.05. MACC overestimates strongly (MB values higher than 0.1) DOD over the area confined by the Caspian Sea, Kazakhstan, Uzbekistan and Turkmenistan in spring, summer and autumn. This area extends over a large part of Iran especially during summer and autumn. Another area where DOD is strongly overestimated by MACC is the area around the so-called Libyan Desert at the triangle between Egypt, Sudan and Libya in winter, summer and mostly autumn. Over the Arabian Sea MACC overestimates DOD throughout the

year, the MB being larger in summer. On the other hand, MACC strongly underestimates (MB values lower than -0.1) DOD over the region of western Sahara during summer and spring and to a lesser extent during autumn and over the Middle East during spring and summer.

The monthly variabilities of MACC and LIVAS DOD and their MBs over the EU domain and over the 9 sub-regions of interest shown in Fig. 4 complement the results discussed above. In general, MACC and LIVAS exhibit similar monthly

variability structures despite the significant biases indicating that MACC captures well the observed seasonality of DOD for all sub-regions. Over the whole EU domain DOD is consistently overestimated by MACC throughout the year, November and June being the months with the highest and lowest MB, respectively (Fig. 4j). In line with the previous paragraph, for areas away from the sources, such as CE, EE, SWE, CM, EM and ATL, MACC overestimates DOD during spring, summer



and autumn and to a lesser extent in winter (Figs. 4a-f). Over those sub-regions, DOD is slightly underestimated by MACC only in one case (in February over EM). The overestimation is stronger over the regions of CE and EE which are far away from the dust sources in the South. In addition, the overestimation is stronger from late spring to early autumn when MACC shows enhanced DOD values contrary to LIVAS. Over SWE, CM, EM and ATL the monthly variability of MACC DOD is

closer to the LIVAS one compared to CE and EE; however, significant biases are observed for spring, summer and autumn. We see here that both MACC and LIVAS depict clearly the difference in the peak period between the Western (summer peak), the Central (transitional region) and the Eastern (spring peak) part of the Mediterranean Basin. Over CWSah MACC underestimates DOD from February to September when dust loadings peak (Fig. 4g) while over ESah MACC overestimates DOD consistently throughout the year (Fig. 4h). Over ME MACC and LIVAS DODs are very close from February to July

while MACC overestimates DOD during the rest of the year (Fig. 4i). In line with the discussion in the previous paragraph DOD peaks in summer over CWSah, in spring over ESah and during spring and summer over ME.

## 3.2 Evaluation of the MACC reanalysis dust profiles

### 3.2.1 Annual dust profiles

In this section, the evaluation of the annual MACC reanalysis profiles is presented taking advantage of the unique ability of

CALIOP/CALIPSO to retrieve dust extinction coefficient profiles. As discussed in Sect. 2.3, the extinction coefficient patterns presented in this work are reported at four 1800-meter layers that cover the first ~9 kilometres of the troposphere (from 1200 to 8400 m above the sea level - a.s.l.). In accordance to Fig. 2, the MACC and LIVAS dust extinction coefficient patterns presented in Fig. 5 are similar over areas characterized by high dust loadings, especially for the first 3 layers (1200-6600 m). The fourth layer (layer 4: 6600-8400 m a.s.l.) is characterized by zero or near zero LIVAS extinction coefficients

everywhere while this is not the case for MACC. MACC overestimates extinction coefficients consistently over the whole EU domain within layer 4 showing that small amounts of dust are always present in MACC even at altitudes up to ~9 km and also over remote oceanic regions. As shown in Fig. 5, within the other 3 layers the overestimations and underestimations from MACC compared to LIVAS are observed at the same areas where MACC overestimates or underestimates DOD (Fig. 2). The absolute MB values are higher in the lowest layer (layer 1: 1200-3000 m a.s.l.) decreasing in layer 2 (3000-4800 m

a.s.l.) and layer 3 (4800-6600 m a.s.l.) which is expected taking into account that dust mostly resides within the first 5 km of the atmosphere in this area (see also Tsikerdekis et al. 2017; Marinou et al., 2017).

In line with Fig. 2, strong overestimation is found over the region situated on the west of the Caspian Sea, the region of eastern Sahara and the Arabian Sea for the first three layers but also over the north-eastern Atlantic Ocean within layer 1. The strong overestimation by MACC within layer 1 over the north-eastern Atlantic Ocean is probably due to the presence of

sea salt aerosols despite the fact that this layer is expected to be higher than the oceanic boundary layer as discussed in detail in Sect. 2.3. On the other hand, MACC underestimates extinction coefficients significantly over the region of central and western Sahara and over the largest part of the Middle East for the first three layers as in the case of DOD (see Fig. 2).

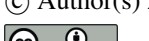



In general, over EU, the MB between the MACC and LIVAS extinction coefficients is 0.006 km$^{-1}$ in layer 1, 0.003 km$^{-1}$ in layer 2 and 3 and 0.002 km$^{-1}$ in layer 4. The corresponding NMB values are 23%, 22%, 105% and 1866%. The correlation coefficient (R) of the linear regression between MACC and LIVAS extinction coefficients is 0.62, 0.75, 0.66 and 0.13 respectively for the four layers. These values along with the mean MACC and LIVAS extinction coefficients, the RMS error, the slope and the intercept of the MACC-LIVAS regression line for each layer for EU and the 9 sub-regions of interest can be found in Table 2. In general, layer 2 exhibits the best correlation between MACC and LIVAS, layers 3 and 1 follow while the correlation is pretty low in layer 4.

Fig. 6 shows the 300-m extinction coefficient profiles from MACC and LIVAS and the corresponding biases for the whole EU domain and the nine sub-regions of interest. As discussed above we focus on altitudes higher than 1 km a.s.l. to avoid as much as possible the interference of sea salt aerosols and hence assume that the MACC natural aerosol extinction coefficients can be similar to dust extinction coefficients. In general, over EU, MACC overestimates extinction coefficients consistently from 1 up to 9 km (Fig. 6j). The overestimation is stable for heights below ~2 km a.s.l. (~0.006 km$^{-1}$) decreases gradually up to ~4 km a.s.l. (~0.002 km$^{-1}$), increases again moderately up to 6 km a.s.l. (~0.003 km$^{-1}$) and finally decreases again, the bias being equal to ~0.002 km$^{-1}$  for heights above 7-8 km a.s.l.. Over CE, EE, SWE and ATL the overestimation by MACC gradually decreases with height until it gets a value of 0.002 km$^{-1}$ at heights above ~7 km a.s.l. (Figs. 6a, b, c and f) while over CM and EM it increases first up to ~2 km a.s.l. and then decreases gradually (Figs. 6d and e). Over CWSah extinction coefficients are consistently underestimated for heights up to ~5.5 km a.s.l. and overestimated thereafter (Fig. 6g). Over ESah MACC overestimates significantly extinction coefficients within the first 4 km a.s.l. with a bias peak at ~2 km a.s.l.. Above ~4 km a.s.l. MACC overestimates extinction coefficients less strongly with a bias peak at ~6 km a.s.l. (Fig. 6h). Over ME MACC underestimates extinction coefficients up until ~2 km a.s.l. and consistently overestimates thereafter (Fig. 6i).

The appearance of non-zero extinction coefficients at heights well above 5 km a.s.l. in the MACC aerosol product, in contrast to ground or satellite-based observations, can be spotted in figures of previous studies (e.g. Fig. 9 in Cuevas et al., 2015 and Fig. 5 in Ansmann et al., 2017). However, there has been no effort to understand the reasons for this situation. According to the discussion in Sect. 2.1.1, this might be due to the way the model deals with the dust distribution in different size bins and dust deposition, vertical transport and mixing. An overestimation of the fine mode dust particles, an underestimation of the dry or wet deposition or a model parameterization that enhances the vertical transport of dust in the atmosphere would justify the existence of particles at heights up to 9 km a.s.l. over the whole EU domain (see Tsikerdekis et al., 2017). In addition, the appearance of non-zero dust extinction coefficients at remote oceanic areas and areas away from the sources could be related to the MACC assimilation procedure. As discussed in Sect. 2.1.1 in detail, an underestimation of the modeled total AOD at each time step over the greater European domain relative to MODIS DT data is possible as ammonium nitrate aerosols, which can affect the AOD directly and indirectly through the absorption of water (Karydis et al., 2016), are not included in the model. In this case, during the assimilation procedure the concentrations of the various aerosol components and consequently dust will be enhanced to match the observed AOD. Undetected aerosol layers by CALIPSO





(see Sect 3.1.1) may also play some role. These factors or a combination of them could be responsible for the consistent MACC overestimation at great heights even over regions such as CWSah and ME where dust extinction coefficients are underestimated by MACC for heights below ~6 km a.s.l. and ~2 km a.s.l., respectively.

## 3.2.2 Seasonal biases between MACC and LIVAS dust profiles

In this section, the seasonal variability of the bias between MACC and LIVAS dust extinction coefficient profiles is discussed. The seasonal patterns of the biases between the MACC and LIVAS extinction coefficients for the four reference layers are presented in Fig. 7. In accordance to Fig. 5 the absolute MB values are higher in layer 1, decreasing in layer 2 and layer 3. In layer 4 MB is consistently positive during all the seasons. MACC overestimates extinction coefficients strongly (MB values higher than 0.02 km$^{-1}$) over the region confined by the Caspian Sea, Kazakhstan, Uzbekistan and Turkmenistan

in spring, summer and autumn and less in winter. The overestimation continues up to layer 4, particularly in spring and summer. Over the eastern Sahara MACC overestimates extinction coefficients mostly in layer 1 throughout the year. Over the Arabian Sea MACC overestimates extinction strongly in layer 1 particularly in winter and autumn. The stronger overestimation in layer 2 appears in spring, summer and autumn while a strong overestimation still appears in layer 3 in summer. Over the north-eastern Atlantic Ocean MACC overestimates extinction strongly within layer 1 throughout the year,

particularly in winter, spring and autumn, the overestimation being much lower in the next 3 layers. On the other hand, MACC strongly underestimates extinction coefficients (MB values lower than -0.02 km$^{-1}$) over western Sahara during summer and spring and to a lesser extent during winter and autumn. Strong underestimation is also seen in spring and summer in layer 2 and in summer in layer 3. Over the Middle East strong underestimations by MACC are seen in spring and summer mostly in layers 1 and 2.

The monthly variability of the bias between MACC and LIVAS 300-m dust extinction coefficient profiles over EU and over the 9 sub-regions of interest is shown in Fig. 8. As it was previously suggested (Sects. 2.3 and 3.2.1), we should focus here on altitudes higher than 1 km a.s.l. to avoid as much as possible the interference of sea salt aerosols in MACC profiles. Over the whole EU domain (Fig. 8j) and for heights lower than ~5 km a.s.l. MACC overestimates extinction coefficients from September to April and underestimates extinction coefficients from May to August. Over ~5 km a.s.l. MACC consistently

overestimates extinction coefficients (MB of ~0.002) with a bias peak in summer (MB of ~0.004) for heights of 6-7 km a.s.l. (Fig. 8a). Over CE, EE and SWE MACC overestimates extinction coefficients consistently throughout the year (Figs. 8a, b and c), the MACC-LIVAS MB decreasing gradually with height. As shown in Fig. 8d, over CM MACC overestimates extinction coefficients at all heights except for the months from June to September at heights lower than ~2 km a.s.l. (for July a small underestimation is seen at a layer located at around 4 km a.s.l.). Over EM we see a similar situation but here the

underestimation period for heights below ~2 km a.s.l. spans from April to September (Fig. 8e). Over ATL MACC overestimates extinction coefficients consistently throughout the year at all the heights except for the summer months when we see a small underestimation at the layer from ~3 to ~5 km a.s.l.. The overestimation is strong at heights below 2-3 km



a.s.l. and gradually decreases (Fig. 8f). Over CWSah MACC underestimates extinction coefficients strongly for heights below 4-5 km a.s.l. from January to October. For the same period MACC overestimates extinction coefficients for heights above 4-5 km a.s.l. while during November and December an overestimation is seen at all height levels (Fig. 8g). Over ESah MACC underestimates extinction coefficients only for heights below ~2 km a.s.l. from January to May and for heights

between 3 and 5 km a.s.l. from March to June. A strong overestimation is seen at heights below ~3 km a.s.l. in November and December. Finally, over ME MACC underestimates extinction coefficients strongly for heights below 2-3 km a.s.l. from January to October while overestimating in November and December. Over ~3 km a.s.l. MACC consistently overestimates throughout the year. Overall, we find that MACC generally overestimates extinction coefficients compared to LIVAS over all the sub-regions except for those who are close to the major dust sources (CWSah, ESah and ME) where MACC

underestimates strongly for heights below ~3-5 km a.s.l. depending on the period of the year.

## 4. Conclusions

In this work, the MACC reanalysis dust product is evaluated over Europe, Northern Africa and Middle East (EU domain) using CALIOP/CALIPSO satellite observations for the period 2007-2012. Specifically, MACC dust optical depth (DOD) data and MACC natural aerosol (dust and sea salt) extinction coefficient profiles at 550 nm are evaluated against DOD and

dust extinction coefficient profiles at 532 nm respectively from the LIVAS pure dust product (Amiridis et al., 2013). As MACC reports only natural extinction coefficients and not dust extinction coefficients a direct evaluation is unfortunately impossible. By focusing on heights above 1 km a.s.l. the influence of sea salt particles (that typically reside at low heights) is diminished and hence it can be assumed that the MACC natural aerosol profile data can be similar to the dust profile data especially over pure continental regions while our results should be considered less robust over the sea and regions close to

the coasts. The main findings of this study are summarized in the following:

- The annual MACC $DOD_{550}$ patterns are close to the LIVAS $DOD_{532}$ ones showing that the reanalysis data are capable of capturing the major dust hot spots in the area. However, MACC overestimates DOD over continental Europe, parts of Turkey and Iran and over the sea (Atlantic Ocean, Mediterranean and Arabian Sea). The overestimation is relatively high

over the region situated on the west of the Caspian Sea and over the eastern Sahara. MACC underestimates DOD significantly over of central and western Sahara and over the largest part of the Middle East. In general MACC overestimates DOD for regions with low dust loadings and underestimates DOD for regions with high dust loadings (DOD exceeds ~0.3). The MB between the MACC and LIVAS DOD is 0.025 over the whole EU, the normalized mean bias (NMB) is ~25%, the root mean squared error (RMS error) is 0.115 and the correlation coefficient (R) of the linear regression (y=0.562x+0.068)

between MACC and LIVAS DOD is 0.76. For DODs lower than ~0.3, the MACC and LIVAS products are characterized by a strong linear correlation with a slope close to 1.




- The MACC reanalysis DODs exhibit a similar spatial variability with the LIVAS DODs during all the seasons. The well documented high dust loadings in summer and spring, especially over Northern Africa and Middle East, are captured by both MACC and LIVAS. For areas more remote from the sources, MACC overestimates DOD during spring, summer and autumn and to a lesser extent in winter (MB values up to 0.05). MACC strongly underestimates (MB values lower than -0.1) DOD

over western Sahara in summer and spring and to a lesser extent in autumn and over the Middle East in spring and summer. In general, MACC and LIVAS exhibit similar monthly structures despite the significant biases. Over the whole EU domain DOD is consistently overestimated by MACC throughout the year. The overestimation is stronger over the regions of CE and EE which are away from the dust sources of the South. Over SWE, CM, EM and ATL the monthly variability of MACC DOD is closer to LIVAS; however, significant biases are observed for spring, summer and autumn. Over CWSah MACC

underestimates DOD from February to September when dust loadings peak while over ESah MACC overestimates DOD throughout the year. Over ME the MACC-LIVAS bias is low from February to July, MACC overestimating DOD during the rest of the year.

- In this work, dust extinction coefficient patterns are reported at four 1800-meter layers (layer 1: 1200-3000 m a.s.l., layer 2:

3000-4800 m a.s.l., layer 3: 4800-6600 m a.s.l. and layer 4: 6600-8400 m a.s.l.). The MACC and LIVAS dust extinction coefficient patterns are similar over areas characterized by high dust loadings, especially for the first 3 layers. Within these layers the overestimations and underestimations from MACC are observed at the same areas where MACC overestimates or underestimates DOD. Layer 4 is characterized by zero or near zero LIVAS extinction coefficients everywhere, MACC overestimating extinction coefficients consistently over the whole EU domain. The MACC-LIVAS extinction coefficient

MB is 0.006 $km^{-1}$ in layer 1, 0.003 $km^{-1}$ in layer 2 and 3 and 0.002 $km^{-1}$ in layer 4. The corresponding NMB values are 23%, 22%, 105% and 1866%. R is 0.62, 0.75, 0.66 and 0.13 respectively for the four layers. In general, layer 2 exhibits the best MACC-LIVAS correlation. Layers 3 and 1 follow while the correlation is low in layer 4. Overall, over EU, MACC overestimates extinction coefficients consistently from 1 up to 9 km a.s.l.. The overestimation is stable for heights below ~2 km a.s.l. (~0.006 $km^{-1}$) decreases gradually up to ~4 km a.s.l. (~0.002 $km^{-1}$), increases again moderately up to 6 km a.s.l.

(~0.003 $km^{-1}$) and finally decreases again, the bias being equal to ~0.002 $km^{-1}$ for heights above 7-8 km a.s.l..

- The absolute MACC-LIVAS MB values are higher in layer 1, decreasing in layer 2 and layer 3. In layer 4 MB is consistently positive during all the seasons. Over the whole EU domain and for heights lower than ~5 km a.s.l. MACC overestimates extinction coefficients from September to April and underestimates extinction coefficients from May to

August. Over ~5 km a.s.l. MACC consistently overestimates extinction coefficients (MB of ~0.002) with a bias peak in summer (MB of ~0.004) for heights of 6-7 km a.s.l.. MACC generally overestimates extinction coefficients compared to LIVAS over the sub-regions away from the major dust sources. On the contrary, over CWSah, ESah and ME MACC underestimates strongly for heights below ~3-5 km a.s.l. depending on the period of the year.




Overall, it is shown in this work that MACC overestimates DOD for regions with low dust loadings and underestimates DOD for regions with high DOD loadings. Non-zero MACC DODs appear over remote areas (away from the source areas in the South) where LIVAS returns zero DODs. In contrast to LIVAS, non-zero MACC dust extinction coefficients can be spotted over the whole EU for heights up to 9 km a.s.l. throughout the year. As discussed above, this could be due to the

model performance and its parameterizations of emissions and other processes, and/or due to the assimilation of $AOD_{550}$ measurements only over dark surfaces (omitting this way the regions where dust is produced) and/or due to a possible enhancement of all the aerosol components (including dust) by the MACC assimilation system over the greater European area. By including a pure dust product such as LIVAS in the assimilation procedure, part of the observed biases would have probably been addressed. Apart from being a potent assimilation tool it is shown here that LIVAS constitutes an ideal

observational dataset for the evaluation of climate model simulations and reanalysis datasets, and the need for more studies towards this direction is acknowledged. It is suggested that dust products from more recent reanalysis projects such as the CAMS interim reanalysis (CAMSiRA) (Flemming et al., 2017), the Modern-Era Retrospective analysis for Research and Applications, Version 2 (MERRA-2) (Gelaro et al., 2017) and the Japanese Reanalysis for Aerosol (JRAero) should be evaluated in a similar way (2-D or 3-D evaluation depending on the availability of dust profile data).

**Acknowledgements**

This research has been financed under the FP7 Programme MarcoPolo (Grand Number 606953, Theme SPA.2013.3.2-01). LIVAS has been financed under the ESA-ESTEC project LIVAS (contract No. 4000104106/11/NL/FF/fk). The researchers from NOA acknowledge the support of the European Research Council under the European Community's Horizon 2020 research and innovation framework program / ERC Grant Agreement 725698 (D-TECT), the European COST Action InDust

(Grand Number CA16202) and the Stavros Niarchos Foundation. CALIPSO data were provided by NASA. We thank the ICARE Data and Services Center (http://www.icare.univ-lille1.fr) for providing access to CALIPSO data used in this study and to their computational center.

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





**Table 1:** MACC $DOD_{550}$ ($\pm 1\sigma$) and LIVAS $DOD_{532}$ ($\pm 1\sigma$) levels, MACC-LIVAS MB, NMB and RMS error, slope and intercept of the MACC-LIVAS linear regression line and the corresponding correlation coefficient R for the Europe - Northern Africa - Middle East (EU) domain and for the 9 sub-regions of interest.

| Region | MACC $DOD_{550}$ | LIVAS $DOD_{532}$ | MB | NMB (%) | RMS error | R | a | b | Obs. |
|---|---|---|---|---|---|---|---|---|---|
| EU | 0.124±0.127 | 0.099±0.173 | 0.025 | 25 | 0.115 | 0.76 | 0.562 | 0.068 | 317892 |
| CE | 0.033±0.023 | 0.012±0.031 | 0.021 | 168 | 0.037 | 0.40 | 0.300 | 0.029 | 14721 |
| EE | 0.042±0.037 | 0.016±0.037 | 0.027 | 169 | 0.047 | 0.45 | 0.445 | 0.035 | 14025 |
| SWE | 0.049±0.047 | 0.026±0.056 | 0.023 | 86 | 0.046 | 0.71 | 0.590 | 0.034 | 5497 |
| CM | 0.067±0.066 | 0.050±0.086 | 0.017 | 34 | 0.061 | 0.74 | 0.565 | 0.039 | 7704 |
| EM | 0.071±0.058 | 0.050±0.081 | 0.022 | 44 | 0.065 | 0.66 | 0.467 | 0.048 | 16062 |
| ATL | 0.070±0.087 | 0.046±0.107 | 0.024 | 52 | 0.058 | 0.87 | 0.713 | 0.037 | 30478 |
| CWSah | 0.176±0.117 | 0.184±0.221 | -0.008 | -4 | 0.161 | 0.71 | 0.376 | 0.107 | 25640 |
| ESah | 0.172±0.091 | 0.123±0.142 | 0.049 | 40 | 0.120 | 0.63 | 0.405 | 0.122 | 15121 |
| ME | 0.246±0.139 | 0.213±0.215 | 0.033 | 16 | 0.163 | 0.67 | 0.432 | 0.154 | 36034 |



**Table 2:** MACC dust extinction coefficient at 550 nm (in km-1) (±1σ) and LIVAS dust extinction coefficient at 532 nm (in km$^{-1}$) (±1σ), MACC-LIVAS MB and RMS error, slope and intercept of the MACC-LIVAS linear regression line and the corresponding correlation coefficient R for the Europe - Northern Africa - Middle East (EU) domain and for the 9 sub-regions of interest for layer 1 (1200-3000 m a.s.l.), layer 2 (3000-4800 m a.s.l.), layer 3 (6800-6600 m a.s.l.) and layer 4 (6600-8400 m a.s.l.).

| Layer | Region | MACC ext$_{550}$ | LIVAS ext$_{532}$ | MB | RMS error | R | a | b |
|---|---|---|---|---|---|---|---|---|
| 1 | EU | 0.030±0.028 | 0.025±0.051 | 0.006 | 0.041 | 0.62 | 0.341 | 0.022 |
| 2 | EU | 0.015±0.019 | 0.012±0.029 | 0.003 | 0.019 | 0.75 | 0.502 | 0.009 |
| 3 | EU | 0.005±0.007 | 0.003±0.009 | 0.003 | 0.008 | 0.66 | 0.515 | 0.004 |
| 4 | EU | 0.002±0.002 | 0.000±0.001 | 0.002 | 0.003 | 0.13 | 0.290 | 0.002 |
| 1 | CE | 0.011±0.010 | 0.003±0.011 | 0.008 | 0.016 | 0.11 | 0.096 | 0.011 |
| 2 | CE | 0.005±0.005 | 0.001±0.004 | 0.004 | 0.006 | 0.44 | 0.497 | 0.004 |
| 3 | CE | 0.002±0.002 | 0.000±0.001 | 0.002 | 0.003 | 0.22 | 0.441 | 0.002 |
| 4 | CE | 0.001±0.002 | 0.000±0.000 | 0.001 | 0.002 | 0.06 | 0.314 | 0.001 |
| 1 | EE | 0.009±0.010 | 0.004±0.011 | 0.005 | 0.013 | 0.37 | 0.322 | 0.008 |
| 2 | EE | 0.004±0.005 | 0.001±0.003 | 0.003 | 0.006 | 0.53 | 0.834 | 0.004 |
| 3 | EE | 0.002±0.002 | 0.000±0.001 | 0.002 | 0.003 | 0.29 | 0.759 | 0.002 |
| 4 | EE | 0.001±0.001 | 0.000±0.000 | 0.001 | 0.002 | 0.06 | 0.177 | 0.001 |
| 1 | SWE | 0.014±0.013 | 0.006±0.017 | 0.008 | 0.018 | 0.44 | 0.333 | 0.012 |
| 2 | SWE | 0.007±0.008 | 0.003±0.010 | 0.004 | 0.008 | 0.72 | 0.572 | 0.005 |
| 3 | SWE | 0.003±0.004 | 0.001±0.004 | 0.002 | 0.004 | 0.61 | 0.587 | 0.003 |
| 4 | SWE | 0.002±0.002 | 0.000±0.001 | 0.002 | 0.003 | 0.14 | 0.219 | 0.002 |
| 1 | CM | 0.017±0.014 | 0.010±0.021 | 0.006 | 0.017 | 0.66 | 0.445 | 0.012 |
| 2 | CM | 0.010±0.012 | 0.007±0.017 | 0.003 | 0.011 | 0.76 | 0.558 | 0.006 |
| 3 | CM | 0.004±0.005 | 0.001±0.006 | 0.002 | 0.005 | 0.66 | 0.542 | 0.003 |
| 4 | CM | 0.002±0.002 | 0.000±0.001 | 0.002 | 0.003 | 0.18 | 0.222 | 0.002 |
| 1 | EM | 0.016±0.014 | 0.011±0.024 | 0.005 | 0.022 | 0.51 | 0.298 | 0.013 |
| 2 | EM | 0.009±0.010 | 0.005±0.013 | 0.004 | 0.010 | 0.71 | 0.572 | 0.006 |
| 3 | EM | 0.003±0.004 | 0.001±0.004 | 0.003 | 0.004 | 0.55 | 0.507 | 0.003 |
| 4 | EM | 0.002±0.001 | 0.000±0.001 | 0.002 | 0.002 | 0.16 | 0.163 | 0.002 |
| 1 | ATL | 0.024±0.022 | 0.009±0.024 | 0.014 | 0.022 | 0.74 | 0.650 | 0.018 |
| 2 | ATL | 0.011±0.016 | 0.007±0.022 | 0.004 | 0.012 | 0.86 | 0.612 | 0.007 |
| 3 | ATL | 0.004±0.006 | 0.002±0.007 | 0.003 | 0.006 | 0.76 | 0.572 | 0.003 |
| 4 | ATL | 0.002±0.002 | 0.000±0.001 | 0.002 | 0.003 | 0.10 | 0.212 | 0.002 |
| 1 | CWSah | 0.037±0.024 | 0.045±0.060 | -0.008 | 0.049 | 0.63 | 0.254 | 0.026 |
| 2 | CWSah | 0.021±0.019 | 0.027±0.040 | -0.006 | 0.029 | 0.77 | 0.371 | 0.011 |
| 3 | CWSah | 0.007±0.007 | 0.007±0.015 | 0.000 | 0.011 | 0.75 | 0.376 | 0.004 |
| 4 | CWSah | 0.003±0.002 | 0.000±0.002 | 0.002 | 0.003 | 0.21 | 0.300 | 0.003 |
| 1 | ESah | 0.036±0.020 | 0.028±0.035 | 0.008 | 0.029 | 0.60 | 0.343 | 0.027 |
| 2 | ESah | 0.019±0.017 | 0.016±0.025 | 0.003 | 0.017 | 0.76 | 0.521 | 0.011 |
| 3 | ESah | 0.006±0.006 | 0.003±0.009 | 0.003 | 0.007 | 0.69 | 0.470 | 0.005 |
| 4 | ESah | 0.003±0.002 | 0.000±0.001 | 0.003 | 0.003 | 0.11 | 0.164 | 0.003 |
| 1 | ME | 0.055±0.031 | 0.059±0.070 | -0.004 | 0.060 | 0.54 | 0.242 | 0.040 |
| 2 | ME | 0.029±0.024 | 0.024±0.033 | 0.005 | 0.024 | 0.69 | 0.502 | 0.017 |
| 3 | ME | 0.010±0.011 | 0.006±0.012 | 0.004 | 0.010 | 0.70 | 0.634 | 0.006 |
| 4 | ME | 0.003±0.003 | 0.000±0.001 | 0.003 | 0.004 | 0.14 | 0.288 | 0.003 |





**Figure 1:** Flowchart with the procedure followed for the evaluation of the columnar (a) and profile (b) MACC reanalysis dust-related datasets using the LIVAS CALIOP/CALIPSO satellite-based dataset.



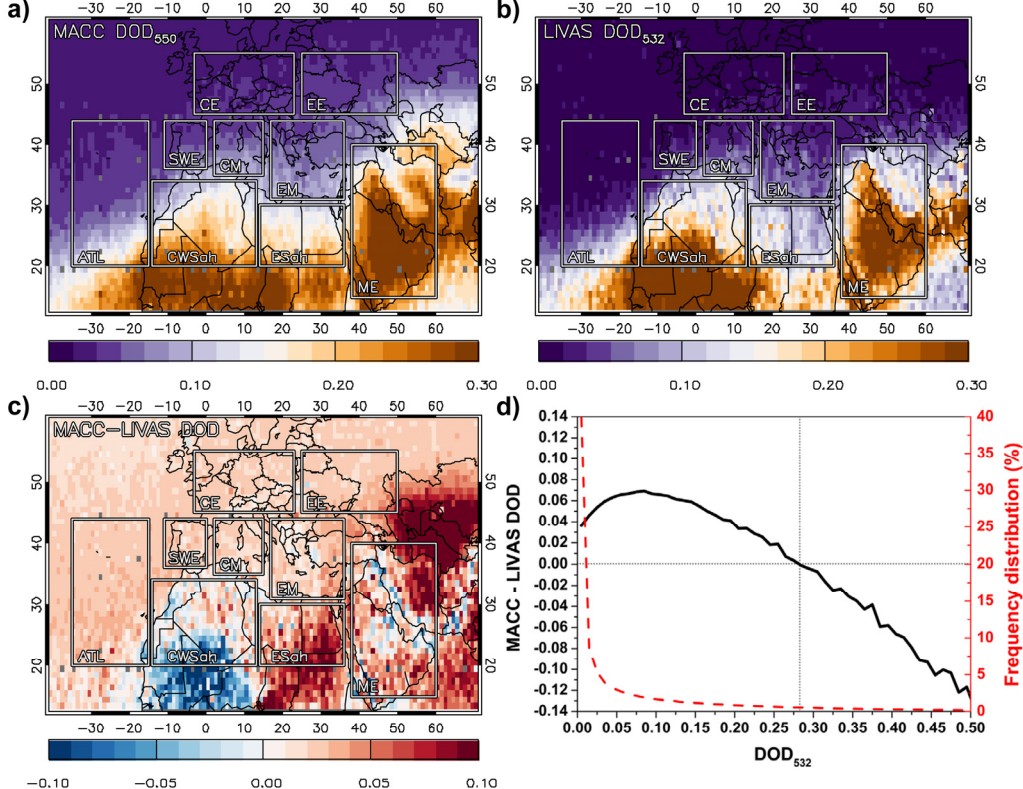

**Figure 2:** Annual patterns of the MACC DOD$_{550}$ (a), the LIVAS DOD$_{532}$ (b) and their mean bias (MB) over the Europe - Northern Africa - Middle East (EU) domain. The nine sub-regions used for the generalization of the results are marked on the maps: Central Europe (CE), Eastern Europe (EE), South-western Europe (SWE), Central Mediterranean (CM), Eastern Mediterranean (EM), Atlantic Ocean (ATL), Central-western Sahara (CWSah), Eastern Sahara (ESah) and Middle East (ME). (d) Plot with the MACC-LIVAS DOD difference relative to the LIVAS DOD$_{532}$ (black line) and the frequency distribution (%) of the LIVAS DOD$_{532}$ values (red dashed line). It is shown that on average the MACC-LIVAS DOD difference turns negative when the LIVAS DOD$_{532}$ exceeds 0.28 while ~90% of the LIVAS DOD$_{532}$ values are below this value.

**Figure 3:** Seasonal patterns of the MACC DOD$_{550}$ (left column), the LIVAS DOD$_{532}$ (middle column) and the MACC-LIVAS mean bias (right column) over the Europe - North Africa - Middle East domain for winter (DJF) (a, b, c), spring (MAM) (d, e, f), summer (JJA) (g, h, i) and autumn (SON) (j, k, l).





**Figure 4:** Monthly variability of the MACC $DOD_{550}$ (orange color) and the LIVAS $DOD_{532}$ (black color) and their mean bias (red color) over the nine sub-regions of interest (a-i) and over the whole Europe - North Africa - Middle East domain (j). Different scale is used for each sub-region so that the differences between the two datasets per sub-region are depicted more efficiently. The monthly variability of DOD for all the sub-regions of interest together for MACC (k) and LIVAS (l) is also presented here to get an insight into the differences in DOD levels over different sub-regions.

**Figure 5:** Patterns of the MACC average dust extinction coefficient at 550 nm (in km$^{-1}$) (left column), the LIVAS average dust extinction coefficient at 532 nm (in km$^{-1}$) (middle column) and the MACC-LIVAS mean bias (right column) over the Europe - North Africa - Middle East domain for layer 1 (1200-3000 m a.s.l.) (a, b, c), layer 2 (3000-4800 m a.s.l.) (d, e, f), layer 3 (6800-6600 m a.s.l.) (g, h, i) and layer 4 (6600-8400 m a.s.l.) (j, k, l).





**Figure 6:** 300-m resolution profiles of the MACC dust extinction coefficient at 550 nm (in km$^{-1}$) (orange color), the LIVAS dust extinction coefficient at 532 nm (in km$^{-1}$) (black color) and their mean bias (red color) over the nine sub-regions of interest (a-i) and over the whole Europe - North Africa - Middle East domain (j). Different scale is used for each sub-region so that the differences between the two datasets per sub-region are depicted more efficiently. The profiles for all the sub-regions of interest together for MACC (k) and LIVAS (l) are also presented here to get an insight into the dust profile differences over different sub-regions.



**Figure 7:** Seasonal patterns (DJF: column 1, MAM: column 2, JJA: column 3 and SON: column 4) of the mean bias between the MACC average dust extinction coefficient at 550 nm (in km$^{-1}$) and the LIVAS average dust extinction coefficient at 532 nm (in km$^{-1}$) over the Europe - North Africa - Middle East domain for layer 1 (a, b, c, d), layer 2 (e, f, g, h), layer 3 (i, j, k, l) and layer 4 (m, n, o, p).



**Figure 8:** Monthly variability of the MACC-LIVAS dust extinction coefficient mean bias profiles (in km$^{-1}$) over the nine sub-regions of interest (a-i) and over the whole Europe - North Africa - Middle East domain (j).

