# Peer review of "A 3-D evaluation of the MACC reanalysis dust product over Europe, Northern Africa and Middle East using CALIOP/CALIPSO dust satellite observations"

_Atmospheric Chemistry and Physics, 2017_

## Referee Comment (RC1) · Anonymous Referee #3 · 21 Feb 2018

General comments

This paper presents an evaluation of the MACC reanalysis dust product by using the pure dust LIVAS database constructed from the space-borne Lidar CALIOP/CALIPSO. Focusing on 3 regions, namely Europe, Northern Africa and Middle East, the evaluation covers both the columnar properties of dust (dust optical depth) and the vertical distribution of the natural aerosols (dust and sea salts) in 4 layers from 1200 m to 8400 m a.s.l. In the Introduction, the authors summarize the different effects of dust and highlight the needs in the dust production and transport forecasts. The second

section is dedicated to the presentation of the datasets (MACC reanalysis and LIVAS CALIOP/CALIPSO) and the methodology. The comparisons are performed with time and space colocated datasets on a monthly granularity. In order to limit the contribution of the sea salts in the natural profiles delivered by the model, the authors focus in 4 separated layers located above the altitude of 1.2 km. The results are presented in the section 3, starting with the columnar dust load. The spatial distribution of dust is well reproduced by the model that captures the major hot spots. Underestimated DODs are found in areas close to the dust sources associated with high dust loads (DOD>0.3), while DODs are overestimated elsewhere, which represent the majority of the cases. The correlation is much higher for cases where DOD<0.3. The seasonality is quite well reproduced by the model which reveals similar annual cycles with the observations in the different regions despite the important bias previously described. The resulting bias over the whole domain presents maximum values in late autumn around 0.03, and a minimum in summer, but still greater than zero. The evaluation of the profiles shows that the model overestimates systematically the extinction in the highest layer over the whole domain. This layer is also associated with a poor correlation. In respect with the columnar loads observations, overestimations are found far from the sources while extinction underestimation are found close to the dust sources. A large overestimation of the extinction in the lowest layer is found in the Atlantic region, due to the presence of sea salts above 1.2 km in the model. The absolute bias tends to decrease with the altitude. The monthly analysis of the profiles reveals similar patterns than for the columnar loads, with greater absolute biases located in the lowest layers.

The paper brings novelty in using original datasets for an extended period and over different regions. The vertical distribution is a key parameter for the aerosols modelling. Such a study, highlighting the differences with observations, is of help for improving the aerosols modelling. The scope, therefore innovative, is well-addressed. The paper is well written, the Figures of high quality, and the English is precise.

I recommend it for publication after the authors respond to the few points listed hereafter.

Specific comments/questions

Following, some specific comments: 3.1.1, p8: different hypothesis are given in order to explain the respective overestimation and underestimation of the dust load in the datasets in the different regions. First, the fraction of fine and coarse particles that could permit to the dust to be transported further from the sources if the dust are considered as too fine in the model. It might be interesting to compare these fractions of fine/coarse mode with some climatological value given by the AERONET retrievals at some key stations (even though this won't say how, in the column, varies this fraction). 3.1.1, p8: another hypothesis given by the authors is the limited sensitivity of CALIOP, as mentioned especially daytime, for detecting small amount of dust. Do you think that the statistics would be significantly different using only night-time Lidar data? 3.2.1: regarding the annual profiles, there is no mention about the associated uncertainty. Is it available from LIVAS product, and what is the confidence in the layer 4 observations (overestimation of the model) when one knows that CALIOP sensitivity is limited when the extinction is small?

Technical corrections

No major technical correction have been found: Figure 3: the three last letters of the subfigures j), k) and l) are missing Figure 6: Hide, similarly to Figure 8 (gray area), below 1.2 km of altitude as well? Or show in both, but this should be consistent Figure 6: The averaging time period should be mentioned in the legend

---

## Referee Comment (RC2) · Anonymous Referee #1 · 18 Apr 2018

The authors present in their paper a 3-D evaluation of the MACC dust product using the EARLINET-optimized dust product from CALIPSO generated from the LIVAS project. They perform a detailed comparison focusing over Europe, Northern Africa and Middle East both for the total dust optical depth as well as for the dust profile product of MACC. The analysis is very detailed and well-presented and the paper is well written. The paper is suitable for publication in ACP after considering some general and specific issues detailed below in my review.

General comment

[Figure]

The structure of section 3 separates the DOD and profile evaluation from MACC and examines separately differences with CALIPSO at annual and seasonal scales. In principle this a reasonable approach. However as it is written and structured, the text has many repetitions accompanied by the same explanations. I would suggest to consider revising the structure of this section focusing eventually on the most interesting regions (or even merge regions in the discussion) and for each region then the authors could compare DOD and profiles at various time scales. This way they will avoid repeating the same discussion in different sections. In addition, in the discussion many differences are attributed to possible modelling issues related to the assimilation or model parametrization in a very generic way, and the text as written lacks justification and seems speculative.

Figures 2, 3 and 5. The color scale used makes the figures hard to read, especially for values close to zero. The authors should consider choosing a different color scale.

Specific comments

Page 4 line 7. What do the authors mean by "if used properly".

Section 2.1 It is not clear how the assimilation of MODIS data is associated with the dust product and the natural aerosol product. Please provide more information here because this info is later used in the discussion of the results.

Section 2.3 (page 5, line 30). Why the authors interpolate the model levels to 399 LIVAS levels and then regrid vertically with 300m resolution instead of first converting LIVAS to the 60 MACC layers and directly average then vertically over the four 1800-meter layers? Why did they choose 1800m? The authors should justify better why they think this way they obtain more robust statistics.

Section 2.3 (page 6, lines 7-15) Are there any estimates how much is the contribution of marine aerosols in the natural aerosols above 1km?

[Figure]

2018.

---

## Author Comment (AC1) · 27 May 2018

**Response to Anonymous Referee #3**

The authors would like to thank Anonymous Referee #3 for his/her comments. Below, please find our response to each one of the referee's comments:

1) 3.1.1, p8: different hypothesis are given in order to explain the respective overestimation and underestimation of the dust load in the datasets in the different regions. First, the fraction of fine and coarse particles that could permit to the dust to be transported further from the sources if the dust are considered as too fine in the model. It might be interesting to compare these fractions of fine/coarse mode with some climatological value given by the AERONET retrievals at some key stations (even though this won't say how, in the column, varies this fraction).

We thank the reviewer for his/her comment. Unfortunately, extinction data are not available from MACC for the single species/bins. Only mixing ratios in different species/bins have been saved. From those it would be indeed possible to calculate the extinction and then the DOD taking into account the optical properties for dust that were used in the MACC model. However, that would require a lot of post-processing of the raw data which might be a task for a future study. This would also fit future developments of the satellite-based dataset used here (coarse/fine mode retrievals).

2) 3.1.1, p8: another hypothesis given by the authors is the limited sensitivity of CALIOP, as mentioned especially daytime, for detecting small amount of dust. Do you think that the statistics would be significantly different using only night-time Lidar data?

We thank the reviewer for giving us the opportunity to clarify this. We are currently not sure if the statistics would be significantly different between day and night datasets as this depends on both the model and the satellite-data.

In LIVAS clear-sky dust product, similar values are observed above Europe for the day and night means. On the contrary, slightly smaller mean seasonal values are observed during daytime above the northern part of Africa and in particular between [20ºN, 30ºN]. For that reason, we might see somehow different statistics between the model and the observations in a day/night inter-comparison.

In addition, the new version 4 of CALIPSO product, with a new enhanced calibration approach, is documented to provide more accurate retrievals. In particular, V4 night-time calibration coefficients coincident with HSRL measurements were found to agree within ~1.6%±2.4% in V4, reduced from 3.6%±2.2% in V3 (Kar et al. 2018). Furthermore, from preliminary studies in our group, we know that the new V4 product includes layers that were undetected in V3 and we are still investigating their contribution in the total dust load.

We acknowledge that a future extension of this work with the new V4 and with daytime / nightime separation could add on this work and provide feedback on which part of the documented differences are due to the model and which due to the satellite.

Reference:

Kar, J., Vaughan, M. A., Lee, K.-P., Tackett, J. L., Avery, M. A., Garnier, A., Getzewich, B. J., Hunt, W. H., Josset, D., Liu, Z., Lucker, P. L., Magill, B., Omar, A. H., Pelon, J., Rogers, R. R., Toth, T. D., Trepte, C. R., Vernier, J.-P., Winker, D. M., and Young, S. A.: CALIPSO lidar calibration at 532 nm: version 4 nighttime algorithm, Atmos. Meas. Tech., 11, 1459-1479, https://doi.org/10.5194/amt-11-1459-2018, 2018.

3) 3.2.1: regarding the annual profiles, there is no mention about the associated uncertainty. Is it available from LIVAS product, and what is the confidence in the layer 4 observations (overestimation of the model) when one knows that CALIOP sensitivity is limited when the extinction is small?

We thank the reviewer for giving us the opportunity to clarify this in the revised manuscript. The associated uncertainty of LIVAS profiles is described in detail in Marinou et al. (2017). We rephrased the document in the end of section 2.2 in order to include the uncertainty of the product.

Regarding the confidence in the layer 4 observations affected from the limited sensitivity of CALIPSO in small extinction layers, we quote the recent publication of Tackett et al. (2018):

*"Several researchers have recently sought to characterize the optical depths of the aerosol layers undetected by CALIOP using collocated observations (Kacenelenbogen et al., 2011; Sheridan et al., 2012; Rogers et al., 2014; Thorsen and Fu, 2015; Toth et al., 2018) or independent retrievals (Winker et al., 2013; Kim et al., 2017). Exactly how these undetected layers affect the level 3 mean extinction is difficult to estimate given that the resulting underestimate depends on the magnitude of missing extinction and the frequency of non-detection. Answering this question is a topic for forthcoming level 3 aerosol product validation".*

Following the reviewer's comment we added a sentence in the document, in the end of section 2.2 stating that the documented bias of LIVAS product (-0.03 in comparison with AERONET and -0.02 in comparison with MODIS) may be attributed to the undetected aerosol layers of CALIPSO.

*"…The correction leads to an $AOD_{532}$ absolute bias of ~-0.03 compared to spatially and temporally collocated AERONET observations above Europe and North Africa while the corresponding biases for the standard CALIPSO product are much higher (~-0.10) (Amiridis et al., 2013). The bias is lower (~-0.02) when compared against spatially and temporally collocated MODIS satellite data.* **This bias may be attributed to the undetected aerosol layers of CALIPSO (Kim et al. 2017).** *In addition, the use of a new methodology for the calculation of the pure dust extinction from dust mixtures and an averaging scheme that includes zero extinction values for the non-dust aerosol types allow for further improvement of the LIVAS pure dust product (Amiridis et al., 2013).* **The uncertainty of the LIVAS pure dust product, it is discussed in detail in Marinou et al. (2017). Overall, the uncertainty of the LIVAS dust seasonal profiles is < 54 % close to the surface and at high latitudes and < 20 % at high altitudes and for latitudes up to 45ºN."**

Reference:

2

Tackett, J. L., Winker, D. M., Getzewich, B. J., Vaughan, M. A., Young, S. A., and Kar, J.: CALIPSO lidar level 3 aerosol profile product: version 3 algorithm design, Atmos. Meas. Tech. Discuss., doi:10.5194/amt-2018-97, in review, 2018.

4) Technical corrections

We thank the reviewer for his suggestions. We have taken into account each one of them. The missing letters have been added to the subfigures in Fig. 3, the gray area has been removed from Fig. 8 in order to be consistent with Fig. 6 and the averaging period has been added in the caption of Fig. 2 stating that *"All the panels in this and the rest of the figures of the manuscript refer to the period 2007-2012.".*

---

## Author Comment (AC2) · 27 May 2018

**Response to Anonymous Referee #1**

The authors would like to thank Anonymous Referee #1 for his/her comments. Below, please find our response to each one of the referee's comments:

1) The structure of section 3 separates the DOD and profile evaluation from MACC and examines separately differences with CALIPSO at annual and seasonal scales. In principle this a reasonable approach. However as it is written and structured, the text has many repetitions accompanied by the same explanations. I would suggest to consider revising the structure of this section focusing eventually on the most interesting regions (or even merge regions in the discussion) and for each region then the authors could compare DOD and profiles at various time scales. This way they will avoid repeating the same discussion in different sections.

We thank the reviewer for his/her comment. We acknowledge that there were some repetitions and hence we did some changes in the text in order to improve this. However, we prefer to keep the basic structure of the manuscript with the discussion on an annual and seasonal basis separately as it is. The structure was thoroughly discussed by the co-authors when preparing the manuscript. We decided to do it this way as it makes it easier for the reader to find the information he/she needs. We first present the results for the annual patterns which is more generic and then go through a seasonal analysis which is more interesting for the more experienced readers in MACC and/or CALIPSO data. The sub-regions were selected not only with geographical criteria (close/away from the dust sources) but also in order to be consistent with previous studies in the area. The use of the specific sub-regions makes our results comparable to results from other studies (e.g. Tsikerdekis et al., 2017 where the LIVAS dataset is used for the evaluation of the RegCM regional climate model dust product) and hence we would prefer to include all of them in the discussion.

2) In addition, in the discussion many differences are attributed to possible modelling issues related to the assimilation or model parametrization in a very generic way, and the text as written lacks justification and seems speculative.

We thank the reviewer for giving us the opportunity to clarify this. The authors have gone through a detailed discussion about the reasons that might affect the bias between the reanalysis (model) and satellite-based data. We agree with the reviewer that one cannot be sure how the assimilation or the different parameterizations might affect the data quantitatively. As mentioned in the text and discussed in Ansmann et al. (2017) "…the uncertainties stemming from the complex parameterizations used by the model make it difficult to reach a solid conclusion about the observed overestimations and underestimations…". A different kind of analysis with simulations with and without assimilation of aerosol data and with the use of different model parameterizations could give some answers. A Monte Carlo approach might be useful in that case. However, this would be a good idea for a future focused study.

3) Figures 2, 3 and 5. The color scale used makes the figures hard to read, especially for values close to zero. The authors should consider choosing a different color scale.

We thank the reviewer for helping us to improve our figures. We updated the color bars by using brighter colors in Figs, 2,3,5 and 7. It is easier for the reader to discriminate the near zero values now.

 4) Page 4 line 7. What do the authors mean by "if used properly".

"Properly" means that one should focus on altitudes higher than 1 km as sea salt particles in the area are mostly accumulated within the marine boundary layer, generally at heights below 1 km. Hence, sea salt is expected to have an impact only at the lower levels of the natural aerosol profiles and one might claim that the natural aerosol profiles are equal to dust profiles. We acknowledge that this phrase might puzzle the reader here and hence we removed it in the revised version of the manuscript.

5) Section 2.1 It is not clear how the assimilation of MODIS data is associated with the dust product and the natural aerosol product. Please provide more information here because this info is later used in the discussion of the results.

A detailed explanation of how the aerosol data assimilation works within MACC is given in the fourth paragraph of Sect. 3.1.1. *"…The control variable of the assimilation is the total aerosol mixing ratio calculated by adding the mixing ratios of all species. The AOD550 is calculated from the single species, summed, integrated and then compared to the observations. Through the 4D-Var assimilation algorithm increments in total aerosol mixing ratio are obtained. Those increments are redistributed to all species proportionally to their fractional contribution to the total mass. It has to be noted that the model does not take into account ammonium nitrate aerosols which represents a large component over the greater European area (Giordano et al., 2015). As a result the model will most of the time underestimate AOD relative to the observations and hence the assimilation system will tend to increase the other aerosol components to give the correct AOD overall. Probably, the system allows the presence of dust even at tiny concentrations and so dust always receives a small contribution during the assimilation even when there should be no dust in the atmosphere…"*

6) Section 2.3 (page 5, line 30). Why the authors interpolate the model levels to 399 LIVAS levels and then regrid vertically with 300m resolution instead of first converting LIVAS to the 60 MACC layers and directly average then vertically over the four 1800-meter layers? Why did they choose 1800m? The authors should justify better why they think this way they obtain more robust statistics.

We thank the reviewer for giving us the opportunity to clarify this. The data from LIVAS are much more detailed as they report an extinction coefficient every 60 m for heights below 20 km. The MACC data characterize much broader vertical layers. The authors decided to bring the model data closer to the observations than doing the opposite. By interpolating the MACC data to the LIVAS levels we preserve the detailed vertical features from LIVAS which would not otherwise be possible. These data are available for future use; however, they are difficult to be used in figures. Therefore, we decided to average on a 300 m basis following Cuevas et al. (2015). In fact the vertical interpolation procedure we followed is similar to the interpolation suggested by Cuevas et al. 2015). To generalize our results we then chose four 1800

m layers. 1800 m allows for covering the first 9 km where this study focuses following also the reasoning of Marinou et al. (2017). The first layer starts from 1200 m and hence we manage to avoid contamination from sea salt particles.

7) Section 2.3 (page 6, lines 7-15) Are there any estimates how much is the contribution of marine aerosols in the natural aerosols above 1km?

We thank the reviewer for his question. It depends mostly on the region. Over the ocean, the contribution is dominant as expected; however, over continental regions, the contribution is nearly zero. We present here three different figures for three different sites from the LIVAS climatology which can be found on http://lidar.space.noa.gr: 8080/livas. As shown in Fig. 1, over the oceanic region, marine aerosols have some contribution above 1km (most of it resides below) while for the other two regions the contribution of marine aerosols is near zero regardless the height. In addition, in Fig. 1 the clean marine component is found to be close to polluted dust component below 1 km. Similar results are shown in Nabat et al. (2013) (fig. 9). For the oceanic region shown here the clean marine extinction above 1200 meters (layer 1 lower boundary) is ~50% of the extinction below this height.

[Figure]

[Figure]

[Figure]